# An Atypical Case of Idiopathic Pulmonary Fibrosis in a Patient from Africa

**DOI:** 10.3390/medicina55030067

**Published:** 2019-03-14

**Authors:** Ida Pesonen, Cristian Ortiz, Giovanni Ferrara

**Affiliations:** 1Department of Respiratory Medicine and Allergy, Section of Respiratory Disease and Allergy, B4:09, Karolinska University Hospital, 171 76 Stockholm, Sweden; giovanni.ferrara@ki.se; 2Respiratory Medicine Unit, Department of Medicine, Solna, Karolinska Institutet, 171 76 Stockholm, Sweden; 3Department of Pathology, Karolinska University Hospital, 171 76 Stockholm, Sweden; cristian.ortiz-villalon@ki.se; 4Department of Oncology-Pathology, Karolinska Institutet, 171 76 Stockholm, Sweden

**Keywords:** idiopathic pulmonary fibrosis, antifibrotic treatment, peribronchiolar metaplasia

## Abstract

A 39 years old African man presented with fatigue, loss of weight and night sweats; radiology showed a possible usual interstitial pneumonia pattern. The patient missed follow-up visits, and presented again after 3 years with productive cough and general illness. Pulmonary function tests showed a decline of FVC compared to a previous investigation. The CT scans showed progression of the interstitial lung disease, and a multidisciplinary conference recommended to proceed with a surgical lung biopsy. Histopathology showed an atypical pattern, with bronchiolar metaplasia. A new multidisciplinary conference made a diagnosis of IPF, and the patient was treated with antifibrotic drugs with a good effect, reaching stability of lung function. This case report highlights the need to improve knowledge and to better characterize rare pulmonary diseases, and especially IPF, among African patients.

## 1. Introduction

Idiopathic pulmonary fibrosis (IPF) is a progressive disease with many suggested etiologies and with an uncertain incidence [1,2]. The diagnosis of IPF is mostly made by radiology if a pattern of usual interstitial pneumomia (UIP) is seen. It has been suggested that IPF incidence is higher in Europe and North America, while it is lower in Asia and South America [3]. The clinical presentation and course of the disease may differ among different countries, thus many genes has been found to be related to IPF [4]. However, very little is known about incidence and presentation of IPF in Africa. This case report illustrates a relatively young African patient, with an atypical presentation, who finally was diagnosed with IPF.

## 2. Case Presentation

A 39-year-old Eritrean male was referred to the Division of Respiratory Medicine and Allergy of the Karolinska University Hospital in Stockholm, Sweden, due to fatigue, fever at nights, loss of weight and appetite, general body pain and interstitial lung abnormalities at the radiology. He had previously suffered from a myocardial infarction in his home country three years before and also had type two diabetes, hypercholesterolemia and hypertension. He was a smoker (15 pack years at the time of referral) and had no specific occupational exposure. The family history of interstitial lung diseases was unknown since the patient had no contact with his family in Africa. He was not aware of any respiratory disease in his family. Physical examination revealed inspiratory and basal crackles. Rheumatoid factor as well as C–reactive protein, hematology laboratory tests and liver function were unremarkable. Computed tomography (CT) showed bilateral, peripheral, reticular changes and ground glass opacities concentrated mostly basally, additionally a five centimeter’s hiatal hernia; the CT-scan was initially identified possible UIP–pattern (Figure 1A). Pulmonary function tests (PFT) showed a vital capacity (VC) of 74 per cent of the predicted level, forced expiratory volume in one second (FEV1) of 74 per cent of the predicted level and a diffusion capacity of carbon monoxide (DlCO) 72 per cent of the predicted level (Figure 2). The patient underwent a bronchoscopy with no macroscopic findings; no infection from common or atypical pathogens were found in the cultures from bronchial samples. Bronchioalveolar lavage (BAL), performed according to standard procedures in the middle lobe, showed very few lymphocytes and a CD4/CD8-ratio of two, other results were also unremarkable. A treatment with proton-pump inhibitors was prescribed, due to the presence of reflux, but the patient did not continue with the prescribed drugs. The patient missed the follow–up visits, and was therefore discharged from the outpatient clinic.

He presented again three years later to our service due to general illness and productive cough. Unfortunately, he was still smoking. Physical examination revealed crackles as previously prescribed but also mild clubbing. A new CT-scan showed a basal, subpleural interstitial pattern with honeycombing and traction bronchiectasis but also dominating ground-glass-opacities in the same areas. Progress compared to the pattern three years earlier was seen with, according to the radiologist, a pattern more compatible with an atypical UIP or a non–specific interstitial pneumonia (NSIP) (Figure 1B). Pulmonary function tests showed an unchanged VC of 79 per cent, FEV1 of 80 per cent but a clearly deteriorated DlCO of 58 per cent of the predicted level (Figure 2). At this time point, the patient was also screened for autoantibodies (including antinuclear antibodies and anti-neutrophil cytoplasmic antibodies) and HIV, which were all negative. The patient went through another bronchoscopy and BAL, again with, unremarkable findings. The case was discussed at a multidisciplinary team conference (MDC) and a decision to go further with a video-assisted thoracoscopic (VATS) biopsy was made.

Two biopsies, one from the right upper lobe and one from the right lower lobe, showed a pattern of interstitial and fibrotic changes with heterogeneous characteristics (Figure 3). A new MDC (with the presence of a skilled pathologist) made a diagnosis of atypical idiopathic pulmonary fibrosis (IPF) and the patient was started on an antifibrotic treatment with pirfenidone.

Since the start of this therapy, the disease had a slower progression compared to the period before the diagnosis (Figure 2). Nevertheless, PFT showed deterioration of lung function over time, with a drop of 19%, 15% and 25% for the VC, FEV1 and DlCO over a time period of seven years. In CT-scan, a clear UIP–pattern developed over time, with traction bronchiectasis, honeycombing and progression of the reticular abnormalities in the lower lobes (Figure 1C,D). The patient was considered clinically relatively stable at the last follow-up visit in autumn 2018.

## 3. Discussion

This clinical case highlights important issues in the diagnosis and treatment of interstitial and rare lung diseases such as IPF, especially in relatively unexplored groups of patients. The diagnosis of IPF may be particularly difficult in patients from countries where little is known about the disease: the presentation of IPF may vary depending on ethnic and genetic backgrounds [4], and the epidemiology of IPF in Africa is still unknown [3]. For example, an epidemiological study from a multi-ethnic county in Paris showed a higher prevalence of IPF among North Africans compared to Europeans and Afro-Caribbeans (26.9 vs. 5.8 and 4.2 per 100,000, respectively) [5]. On the other hand, a study in the U.S. comparing characteristics between different ethnic groups showed that Whites were more likely to have the diagnosis IPF on their death certificate compared to Blacks while Blacks were more likely to die at a younger age and with pulmonary hypertension [6]. Although these findings are not conclusive, they suggest different presentations of the disease in different ethnic groups. Furthermore, the available recommendations [1,2] which are even used in African countries [7], are based on studies performed in Europe and North America.

Our patient presented at a younger age, affected already by other comorbidities, and with a radiological pattern of possible UIP. The histopathology showed an atypical UIP pattern with peribronchiolar metaplasia (described by Fukuoka et al. [8]), and an aggressive approach and a careful multidisciplinary evaluation were required to reach a diagnosis of IPF. The patient was therefore treated with antifibrotics, achieving stability of lung function. After 10 years, the radiological CT pattern was clearly compatible with IPF.

The patient’s lung function results were computed using the Swedish reference values; again, very little is known about lung function reference values in African populations, and this question has only recently been explored in Sub–Saharan African countries like Nigeria [9]: we may have underestimated the severity of the disease in our patient, due to the lack of appropriate reference values.

## 4. Conclusions

This case report highlights the lack of solid evidence and the need for large studies to characterize the presentation and course of interstitial and rare lung diseases among African patients.

An aggressive approach with multidisciplinary evaluation by expert physicians may be necessary to depict similarities and differences of disease such as IPF, in comparison to what is described in the international literature.

## Figures and Tables

**Figure 1 medicina-55-00067-f001:**
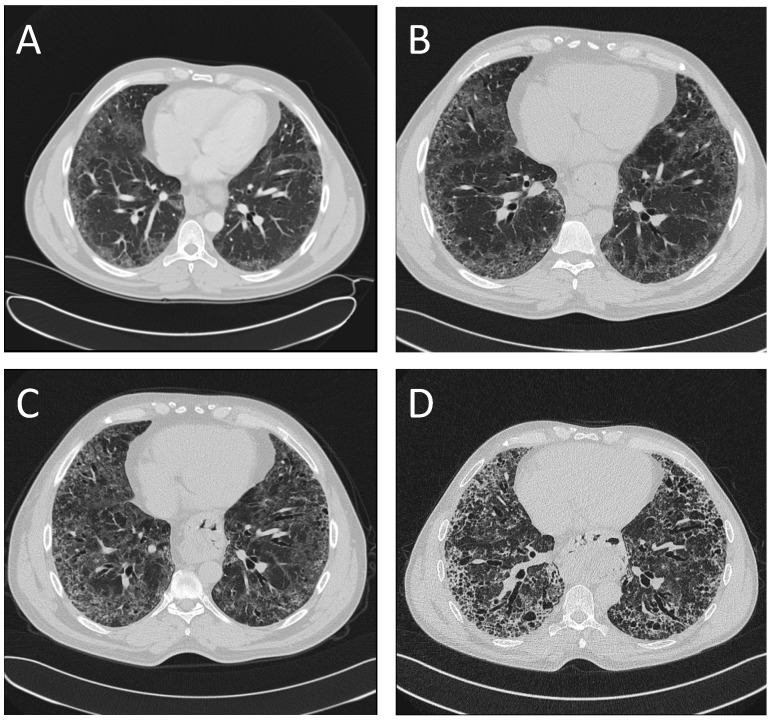
Evolution of the radiological pattern over time, showing basal bilateral reticular changes and ground glass peripheral infiltrates at the presentation of the disease (**A**), development of traction bronchioloectasis and clear honeycombing ((**B**), scans after 3 and (**C**), scans after 6 years), and finally a clear usual interstitial pneumonia (UIP) pattern after 10 years (**D**).

**Figure 2 medicina-55-00067-f002:**
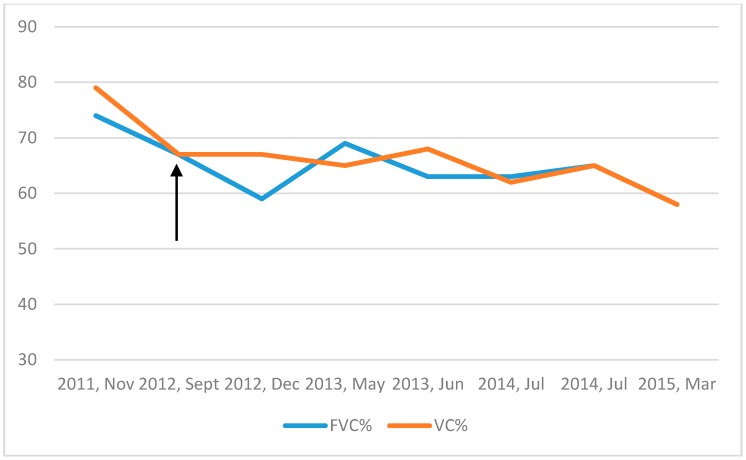
Forced vital capacity, per cent of predicted (FVC%) and vital capacity, per cent of predicted (VC%), showed stable values after the start of antifibrotic treatment (arrow).

**Figure 3 medicina-55-00067-f003:**
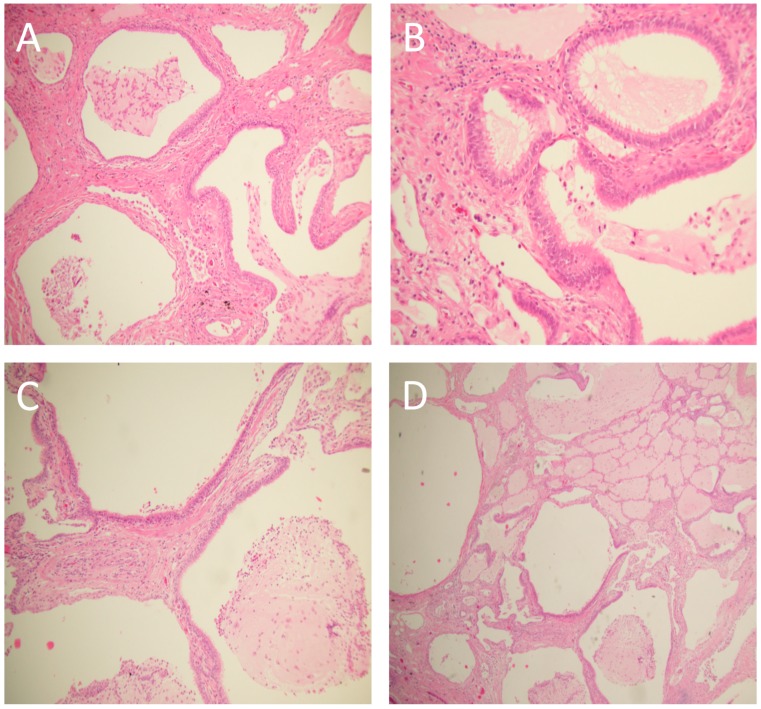
Lung biopsy of the patient. Hematoxylin and eosin-stained (H&E) images of peribronchiolar metaplasia: (**A**) H&E 10×, showing peribronchiolar proliferation of respiratory-bronchial epithelium lining thickened–fibrotic alveolar walls. (**B**,**C**) H&E 20×, at higher magnification the metaplastic epithelium consists of columnar-cuboidal ciliated cells without atypia and absence of goblet cells. (**D**) H&E 10×, peribronchiolar metaplasia associated with a probable usual interstitial pneumonia (UIP)/idiopathic pulmonary fibrosis (IPF) showing heterogeneous interstitial fibrosis with honeycomb, associated with nonfibrotic areas of normal lung parenchyma.

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
