# Peer review of "An Atypical Case of Idiopathic Pulmonary Fibrosis in a Patient from Africa"

_medicina, 2019, doi:10.3390/medicina55030067_

Round 1

Reviewer 1 Report

Thank you for the revised version of the manuscript and I agree with it.

I have not any additional comments.

Reviewer 2 Report

The Authors have made satisfactory amendments to the manuscript.

I believe that the paper is now suitable for publication in "Medicina"

Reviewer 3 Report

An interesting manuscript in its field, I have no comments

This manuscript is a resubmission of an earlier submission. The following is a list of the peer review reports and author responses from that submission.

Round 1

Reviewer 1 Report

This manuscript presents a case report involving a young man with an interstitial lung disease. Unfortunately the case does little to contribute new knowledge and is of very little added value to the literature. From the presentation, it would appear that a number of potential contributing etiologies were either not considered or were not presented such as a telomere mutation or significant GI pathology. In fact, the CT shows considerably abnormal esophagus and raises concern to me that a major, modifiable reflux etiology has been missed in this case.

The writing itself is also not crisp and considerable English language editing would be needed.

Reviewer 2 Report

Major comments

This is a report which draws attention to a matter of significant importance in the area of Idiopathic Pulmonary Fibrosis, namely heterogeneity of clinical syndrome and variability of  histopathologic findings, according to patients’ ethnic and genetic characteristics.

The goal of the report was to describe clinico-radiological characteristics and examine efficacy of anti-fibrotic agents in an atypical IPF case of an African patient.

Results confirm data from other centres, expert in the field, reporting possible differences in clinico-radiological characteristics and the efficacy of anti-fibrotic agents between “atypical” and “typical” form of IPF.

In the opinion of the reviewer, this case could be of interest for the readers of “Medicina”, because it underscores the unexpected, favourable clinical outcome of a case otherwise resulting in a progressive and irreversible respiratory decline.

Minor comments

English language is not always at the expected level and should  undergo a mother tongue review.

Reviewer 3 Report

They describe both diagnosis, UIP and IPF in Case presentation (line 32) but not enough in introduction (line 24).

Figure 2. There is no data of the duration disease course on the Figure 2.

Discussion and Conclusions: pointed out more what are differences in African and Cacasian in the disease.